# *Arabidopsis* *ERF012* Is a Versatile Regulator of Plant Growth, Development and Abiotic Stress Responses

**DOI:** 10.3390/ijms23126841

**Published:** 2022-06-20

**Authors:** Yupu Huang, Ling Liu, Haitao Hu, Ning Tang, Lei Shi, Fangsen Xu, Sheliang Wang

**Affiliations:** 1National Key Laboratory of Crop Genetic Improvement, Huazhong Agricultural University, Wuhan 430070, China; 40060065@henu.edu.cn (Y.H.); hzaulingliu@webmail.hzau.edu.cn (L.L.); leish@mail.hzau.edu.cn (L.S.); fangsenxu@mail.hzau.edu.cn (F.X.); 2Microelement Research Center, College of Resources & Environment, Huazhong Agricultural University, Wuhan 430070, China; 3State Key Laboratory of Crop Stress Adaptation and Improvement, Henan University, Kaifeng 475001, China; 104753211796@henu.edu.cn (H.H.); ningtang@henu.edu.cn (N.T.)

**Keywords:** *ERF012*, plant growth and development, phytohormones, abiotic stress response

## Abstract

The AP2/ERF transcription factors are widely involved in the regulation of plant growth, development and stress responses. *Arabidopsis ERF012* is differentially responsive to various stresses; however, its potential regulatory role remains elusive. Here, we show that *ERF012* is predominantly expressed in the vascular bundles, lateral root primordium and vein branch points. *ERF012* overexpression inhibits root growth, whereas it promotes root hair development and leaf senescence. In particular, *ERF012* may downregulate its target genes *AtC4H* and *At4CL1*, key players in phenylpropanoid metabolism and cell wall formation, to hinder auxin accumulation and thereby impacting root growth and leaf senescence. Consistent with this, exogenous IAA application effectively relieves the effect of *ERF012* overexpression on root growth and leaf senescence. Meanwhile, *ERF012* presumably activates ethylene biosynthesis to promote root hair development, considering that the *ERF012*-mediated root hair development can be suppressed by the ethylene biosynthetic inhibitor. In addition, *ERF012* overexpression displays positive and negative effects on low- and high-temperature responses, respectively, while conferring plant resistance to drought, salinity and heavy metal stresses. Taken together, this study provides a comprehensive evaluation of the functional versatility of *ERF012* in plant growth, development and abiotic stress responses.

## 1. Introduction

Transcription factors (TFs), which are central to gene expression regulation, play critical roles in regulating major developmental processes in the plant life cycle, including seed germination, seedling growth, tissue morphogenesis, reproduction and senescence. Meanwhile, accumulating evidence implicates the importance of various TFs, such as NAC (no apical meristem (nam), *arabidopsis* transcription activation factor (ataf) and cup-shaped cotyledon (cuc)), WRKY, MYB, bHLH (basic helix–loop–helix), bZIP (basic-region leucine zipper) and ERF (ethylene-responsive element binding), in mediating plant growth, development and responses to environmental stresses [1]. For example, the MYC2 TF, a master regulator of the JA signaling pathway, is essentially involved in plant growth, development and defense responses to biotic and abiotic stresses [2,3,4].

The APETALA2/ethylene-responsive factor (AP2/ERF) superfamily widely exists in the plant kingdom, consisting of AP2 (APETALA2), DREB (dehydration-responsive element binding), ERF, RAV (related to ABI3/VP) and Soloist (few unclassified factors) subfamilies [5,6,7,8] based on the number of AP2-conserved domains. In recent years, the roles of ERF TFs in plant growth, development and stress adaptability have become increasingly prominent including, but not limited to, somatic embryogenesis [9], plant root growth [10], internode and shoot elongation [11,12], fruit ripening [13,14], secondary metabolism [15,16], submergence or hypoxia stress [11], heavy metal stress [17], drought stress [10,18,19], high salinity [15,20] and cold stress [16]. For instance, OsERF3 was identified as an important player in cytokinin signaling-mediated crown root development in rice [21]. OsERF3 induces a cytokinin type-A response regulator (*RR*) gene, *RR2*, to regulate crown root initiation, whereas it represses *RR2* expression during crown root elongation through interacting with WOX11, a key regulator in crown root development. In *Arabidopsis*, the AP2/ERF superfamily has 147 members, of which the ERF subfamily accounts for 122 genes [6]. As a model plant, studies of AP2/ERF TFs in *Arabidopsis* provide the greatest amount of basic knowledge for understanding the implication of AP2/ERF TFs in plant life. A good example is the dose-dependent regulation of *Arabidopsis* root development mediated by PLETHORA (PLT) proteins (AP2 TFs) [22]. A clade of four PLT homologs creates a protein gradient distribution along the root apical meristem (RAM) with maxima in the stem cell niche (SCN). A high level is necessary for SCN maintenance and a low level is necessary for cell differentiation [22]. The *Arabidopsis* AP2/ERF TF has been successfully applied in crop engineering for developing sustainable agriculture. For instance, *Arabidopsis HARDY* (*HRD*), encoding an AP2/ERF-like TF, confers drought and salt stress resistances in rice and *Trifolium alexandrinum* L. (an important type of forage crop), due to the reduction of transpiration and sodium uptake [23,24].

The *Arabidopsis ERF012/DREB26* gene encodes a nuclear transcription factor and is differently responsive to NaCl, freezing, heat and drought stresses, as well as to jasmonate (JA) and salicylic acid (SA), whereas the biological function of *ERF012* is poorly understood [25]. Here, we performed a systematic analysis of the *ERF012* expression profile and evaluated its involvement in plant growth and development by establishing null mutants and overexpressing plants. In addition, we investigated the potential roles of *ERF012* in plant resistance to drought, salinity or heavy metals stresses. Our findings highlight the functional versatility of the *ERF012* transcription factor in *Arabidopsis* growth, development and abiotic stress responses.

## 2. Results

### 2.1. Tissue-Specific Expression of ERF012 in Arabidopsis

To understand the tissue-specific expression of *ERF012*, we performed qRT-PCR analysis of *ERFF012* transcription levels in *Arabidopsis* tissues. Strong expression levels were detected in the root, leaves, stem and pod, whereas they were significantly reduced in the bud and flower (Figure 1A). To obtain a high-resolution image of the *ERF012* expression in tissues, the p*ERF012*:GUS (β-glucuronidase) transgenic lines were established (Figure 1B). Based on the GUS staining, the tissue-specificity was detailed. The whole cotyledon was stained, showing that true leaves possessed margin labeling and scattered spots (Figure 1C,D). Interestingly, these spots were specifically localized on the vein branch points. In the root, no GUS staining was observed in the primary and lateral root tip (Figure 1H,I). However, the mature regions, especially the lateral emerged and emerging cells that included the lateral root primordium and vascular cells, had elevated expression (Figure 1E–G). These findings were contrary to a previous report [25] that only strong GUS activity was observed in the cotyledon. We further performed a transactivation assay of *ERF012* in the *Arabidopsis* protoplast. Expression of GAL4-ERF012 activated GAL4-LUC activity largely compared with the GAL4-BD control (Figure 1J), demonstrating that *ERF012* can regulate downstream gene transcription. These results indicate that the *ERF012* transcription factor may play an important role in the growth and development of *Arabidopsis*.

### 2.2. ERF012 Regulates Plant Growth and Development

To explore the effect of *ERF012* on plant growth and development, the *ERF012*-overexpressing (OE) transgenic lines were established (Figure 2E) by transforming a 35S:*ERF012* plasmid in Col-0. In parallel, the *e**rf012* mutants were established using a gene-editing system (CRISPR-Cas9) (Figure 2F). Under normal conditions (1/2 MS medium), Col-0 and *erf102* mutant lines developed comparable primary root lengths and fresh weights, whereas the primary lengths and fresh weights of *ERF012*-OE plants were reduced by 58–68% and 50–70% compared to Col-0, respectively (Figure 2A–C). As shown in Figure 2A,D, *ERF012*-OE lines obviously had a lower lateral root number than Col-0 and *erf012* mutants. We noticed that the root tips of *ERF012*-OE plants were significantly shorter than that of Col-0 plants, including the meristem zone (MZ) and elongation zone (EZ) (Figure 3A,B). Microscopic statistics showed that cell numbers, but not the cell lengths, of MZ and EZ in *ERF012*-OE roots were less than the Col-0 roots (Figure 3C,D). These results suggest that *ERF012* expression might downregulate cell division. To this end, we compared the cyclin-dependent protein kinase (CYCB1;1) activity in Col-0 and *ERF012*-OE backgrounds. CYCB1;1 functions as an effector of growth control at G2/M, thus indicating the cell division rate. We speculated that *CYCB1;1* has higher activity in Col-0 than in *ERF012*-OE plants (Figure 3E). As expected, p*CYCB1;1*:GUS showed stronger expression and activity in the meristem zone of Col-0 than in that of *ERF012*-OE plants (Figure 3F).

### 2.3. ERF012 Downregulates Gene Expression of Cell Wall Formation and Phenylpropanoid Metabolism Pathway

To interpret the growth alterations of *ERF012*-OE plants, we performed the interaction gene analysis of *ERF012* (https://bar.utoronto.ca/eplant/ accessed on 1 April 2022). Many genes were predicted to be the targets of *ERF012* (Appendix A). A part of cell wall formation involves genes such as At1g27440, AT2G34710, AT3G18660, AT5G15630 and AT5G17420 encoding the cell wall formation-related proteins [26]. In addition, the upstream regulatory genes of cell wall formation, At2g30490 and At1g51680, which are involved in the phenylpropanoid metabolism pathway, were listed in the target candidates. At2g30490 and At1g51680 encode a cinnamate 4-hydroxylase (AtC4H) [27] and a 4-coumarate:coenzyme A ligase (At4CL1) [28], respectively, and AtC4H1 directly functions upstream of At4CL1. All predicted targets were downregulated in the *ERF012*-OE plants (Appendix A). The phenylpropanoid metabolism makes an important contribution to the coordination of plant development and plant–environment interaction via cell wall formation containing diverse metabolic routes; thus, the disruption of phenylpropanoid metabolism severely limits plant growth, development and stress responses [29]. Compared with Col-0, the expression levels of *AtC4H* in the shoot and root of *ERF012*-OE plants and *At4CL1* in the shoot of *ERF012*-OE plants were significantly reduced (Figure 4A), demonstrating that *ERF012* overexpression disrupts the phenylpropanoid metabolism. To validate the interaction of *ERF012* with *AtC4H* and *At4CL1*, a yeast one-hybrid assay was conducted. The results showed that both promoters of *AtC4H* and *At4CL1* can directly interact with the *ERF012* protein (Figure 4B). These results suggest that *ERF012* expression downregulate gene expression in cell wall formation and the phenylpropanoid metabolism pathway, which might lead to limited growth.

### 2.4. Auxin and Ethylene Involvement in the Root Growth and Lateral Root Formation in ERF012-OE Plants

Loss of lateral roots was obvious in the *ERF012*-OE plants (Figure 2A). It was reported that the *Atc4h/ref3* mutant reduced lateral roots because blocking of the AtC4H1 function largely hinders the accumulation of lignin and causes slow, phloem-mediated auxin transport [30]. Indeed, the auxin level in the root tip of *ERF012*-OE plants was lower than the Col-0, evidenced by DR5:GFP signals (Figure 4C). Plant hormone auxin is a key factor that controls lateral root formation [31,32,33] due to the auxin-dependent signaling. To investigate whether auxin is involved in lateral root formation in *ERF012*-OE lines, exogenous IAA was used to treat Col-0 and *ERF012*-OE lines under normal growth conditions. Five-day-old seedlings were transferred to normal growth conditions supplemented with 0.1 μM IAA for a further 6 days of growth. With the IAA addition, the lateral roots were formed in large quantities in *ERF012*-OE lines (Figure 4D,E). This result demonstrates that *ERF012* expression limits the formation of lateral roots.

Short root tips and high root hair density are the typical characteristics of excess ethylene in plant roots. Compared to Col-0, the roots of *ERF012*-OE plants developed shorter root tips and dense root hair (Figure 5A), indicating the high ethylene level in *ERF012*-OE roots. To investigate whether ethylene is involved in the root growth of *ERF012*-OE plants, we compared the expression levels of the ethylene precursor and biosynthetic gene 1-aminocyclopropane-1-carboxylic acid (*ACC*) synthase (*ACS*) between Col-0 and *ERF012*-OE plants. Two *ERF012*-OE lines had significantly higher expression levels of *ACS7* and *ACS11* than in Col-0 (Figure 5B), suggesting ethylene accumulation in *ERF012*-OE lines. We further used an ethylene biosynthesis inhibitor to estimate the involvement of ethylene in the root growth of *ERF012*-OE lines. AVG treatment largely improved the root growth of *ERF012*-OE lines (Figure 5C). The root tips were elongated and root hair density was dramatically reduced in the AVG-treated *ERF012*-OE lines (Figure 5B–D). These results demonstrate that *ERF012* expression enhances ethylene biosynthesis, leading to root growth inhibition.

### 2.5. ERF012 Responds Quickly to Temperature Dynamics

To estimate the responses of *ERF012* to environmental stresses, temperature dynamics were first tested. Col-0 seedlings grown under normal conditions (22 °C) were transferred to high temperatures (37 °C) or low temperatures (4 °C) for several hours. Time-course qRT-PCR analysis of *ERF012* expression in the shoots and roots was performed. The rapid responses (within 2 h) to dynamics were observed for Col-0 shoots, whereas no significant changes occurred in the roots (Figure 6A,B). The consistent results were confirmed by p*ERF012*:GUS lines treated at different temperatures (Figure 6C), suggesting that *ERF012* plays a critical role in shoots under temperature dynamics. To validate the involvement of *ERF012* expression in shoot response to temperature dynamics, Col-0, *erf012* mutants and *ERF012*-OE lines were phenotypically compared at different temperatures. At normal conditions (22 °C), *ERF012*-OE shoots had lower chlorophyll content than the Col-0 and *erf012* mutants, corresponding to visible chlorosis (Figure 6D,E). High temperature (37 °C) largely reduced the chlorophyll content in all plants with chlorosis, although the *ERF012*-OE lines still had the lowest levels (Figure 6D,E). Interestingly, the chlorosis of *ERF012*-OE lines at 22 °C and all plants at 37 °C can be lightened largely by the IAA supply (Figure 7A,B). In contrast, the low-temperature treatment (4 °C) did render growth retardation (Figure 6D) and the *erf012* mutants accumulated the highest anthocyanin content and *ERF012*-OE lines accumulated the lowest anthocyanin content (Figure 6D,F). Anthocyanin accumulation is usually an adaptation strategy of plants to low temperatures [34]. Thus, *ERF012* expression disturbed anthocyanin biosynthesis, leading to reduced low-temperature tolerance. These results indicate that *ERF012* expression negatively regulates Arabidopsis adaptation of temperature dynamics.

### 2.6. ERF012 Overexpression Reduces Plant Sensitivity to Abiotic Stresses

To further dissect the possible function of *ERF012* in plant responses to various abiotic stresses, we performed a phenotypical evaluation among Col-0, *erf012* mutants and *ERF012*-OE lines under various stresses including drought (100 mM mannitol), salinity (100 mM NaCl) and arsenite (10 μM NaAsO_2_) and cadmium (15 μM CdSO_4_) toxicity. Drought stimulation limited plant growth to a large extent (Figure 8A–C). Compared with normal conditions (CK), the relative root lengths (RRLs) and relative shoot fresh weights (RSFWs) of erf012 mutants were lower and greater, respectively, whereas the RRLs and RSFWs of ERF012-OE plants did not alter (Figure 8G), suggesting that *ERF012* expression is not sensitive to drought stress. Consistent with this, expression levels of drought stress-responsive genes [35,36,37] were comparable in Col-0 and *ERF012*-OE plants (Appendix A). Except that the #7 *ERF012*-OE line showed worse growth, a similar result was observed due to the salt stress, including the plant growth changes and gene responses to salt stress [38] (Figure 8A–D,G and Appendix A). Cd toxicity was indicated by the shoot chlorosis in all plants with a reduction of fresh weights and primary root length (Figure 8A,E). Compared with normal conditions (CK), the RSFWs and the RRLs of *ERF012*-OE plants were greater and lower, respectively (Figure 8G). Both upregulation of *NRAMP3* and *NRAMP4* and the downregulation of *GSH2* and *PCS2* [39] in *ERF012*-OE plants would reduce the ability of Cd tolerance in the root (Appendix A). Intriguingly, stress largely caused growth retardation of Col-0 and *erf012* plants, whereas the *ERF012*-OE plants showed tolerance accompanied by RRLs and RSFWs (Figure 8A,F,G), suggesting that *ERF012* overexpression reduces the plant sensitivity to As toxicity. The downregulation of NIP1;1, an arsenite channel [40], might contribute to the As tolerance of *ERF012*-OE plants (Appendix A). In summary, because of the strong inhibitory effect of *ERF012* overexpression on plant growth and development at normal conditions, the low sensitivity of *ERF012*-OE plants to various abiotic stress was evident.

## 3. Discussion

The AP2/ERF superfamily is one of the biggest TF families in higher plants and regulates various biological processes of plant growth and development, as well as the response to biotic and abiotic stresses. Most proteins in this superfamily, characterized by a single AP2 domain and introns, are assigned to the ERF family, whereas the proteins characterized by a tandem repetition of two AP2 domains and a small number of proteins with a single AP2 domain are assigned to the AP2 subfamily [6]. It was proposed that AP2 TFs are primarily involved in developmental programs, whereas ERF TFs are mainly responsible for environmental stimuli or hormones [8].

A general DNA element A/GCCGAC usually exists in many ABA-, drought- and cold-responsive genes and can be bound by the DREB proteins. For example, the one AP2 domain containing the CBF1/DREB1B TF was demonstrated to bind to A/GCCGAC in gel-shift assays and showed rapid cold response within 2 h, but did not change appreciably at 24 h [41]. The *ERF012* protein also contains one AP2 domain and belongs to the group II, A5 subfamily corresponding to the ERF subfamily in Nakano’s classification [6] and the dehydration-responsive element binding proteins (DREBs) group in Sakuma’s classification [42]. It has been shown that *ERF012* gene expression in the root of the 12-day-old seedlings was not responsive to the temperature dynamics [25], which is consistent with our observation (Figure 6). However, rapid responses for *ERF012* were detected in seedling shoots within 2 h of low-temperature (4 °C) or high-temperature (37 °C) treatments (Figure 6A,B). Thus, *ERF012* and CBF1/DREB1B have a similar response to low temperature, suggesting that both might play a role in cold stress adaptation. Indeed, the overexpression of *CBF1* in *Arabidopsis* induces the expression of several COR (cold-regulated) genes and increases the freezing tolerance of plants [43]. In agreement with the function of CBF1, the higher anthocyanin content in *ERF012*-OE plants and lower anthocyanin content in *erf012* mutants and Col-0 plants suggest that the expression of *ERF012* plays a positive role in cold adaptability. By contrast, at 22 °C and 37 °C, the *ERF012*-OE plants showed severe chlorosis accompanied by reduced chlorophyll content as compared with Col-0 and *erf012* plants (Figure 6 and Figure 7), suggesting that *ERF012* plays a negative role in photosynthesis at normal-temperature and high-temperature conditions and promoting the leaf senescence process. Leaf senescence represents the functional transition from nutrient assimilation to nutrient remobilization, a process involving complex genetic programs caused by various stresses such as phytohormone dynamics, drought, heat, heavy metal and nutrient deficiency, etc. The *ERF012*-OE plants showed leaf senescence under normal conditions (Figure 6E), which may be caused by phytohormones or nutrient deficiency. The underdeveloped roots of *ERF012*-OE plants would significantly reduce nutrient uptake such as of magnesium (Mg) and iron (Fe). However, the Mg concentration in the shoots of *ERF012*-OE plants was higher than in that of Col-0, and no different Fe concentrations were observed in the shoots of both plants (Appendix A), excluding their contribution to leaf senescence of *ERF012*-OE plants. It has been reported that elevated auxin levels, by expressing an auxin biosynthetic enzyme called the YUCCA6, which mutation can significantly delay the leaf senescence [44]. Indeed, exogenous application of IAA largely delayed the leaf senescence and accompanied the elevation of chlorophyll content in all plants at 22 °C and 37 °C conditions (Figure 7), although it was still lower in the *ERF012*-OE plants. The increased root hair and the enhanced expression of ethylene biosynthetic genes *ACS7* and *ACS11* in *ERF012*-OE plants (Figure 5A–C) indicate that there is more ethylene in *ERF012*-OE plants. This does not exclude the ethylene, which plays an important role in the onset of leaf senescence. For example, overexpression of *AtERF4* positively regulates leaf senescence and loss of function of *AtERF4* delays leaf senescence in an EIN3-dependent manner [45,46]. These results indicate that expression of *ERF012* might repress endogenous auxin accumulation, which at least partially contributes to leaf senescence.

The auxin-deficient growth was also evident in the *ERF012*-OE roots with few lateral roots (Figure 2 and Figure 4). The root system is crucial for plant growth and the lateral roots contribute, essentially, to ensuring anchorage acquisition of water and nutrients from the soil. In *Arabidopsis*, lateral roots are initiated at the protoxylem poles, called founder cells, by high levels of auxin stimuli. A series of auxin components have been widely identified, for example, in a study that evaluated auxin and found it causes the degradation of IAA/AUX and increases the auxin response factors, ARF7 and ARF19, which directly regulate the auxin-mediated transcription of lateral organ boundaries-domain16/asymmetric leaves2-like18 (LBD16/ASL18) and/or LBD29/ASL16 to activate lateral root formation in roots [47]. ERF TFs regulate auxin signaling or are regulated by auxin in lateral root formation. *Arabidopsis* ERF109 binds directly to the promoters of auxin biosynthesis, *ASA1* and *YUC2*, to evaluate the auxin level in lateral root primordium [48] and auxin induces degradation of the ERF13 and releases fatty acid elongase KCS16 to initiate lateral root emergence [49]. Exogenous IAA applications largely restored the lateral root formation (Figure 4D), suggesting that *ERF012* expression represses auxin accumulation in roots as validated by the DR5:GFP signals between Col-0 and *ERF012*-OE roots (Figure 4C). The auxin reduction in *ERF012*-OE roots may also be attributed to the disturbed phenylpropanoid metabolism pathway. In *ERF012*-OE plants, the key enzyme gene *AtC4H1* of the phenylpropanoid metabolism pathway was downregulated and directly interacted with the *ERF012* protein (Figure 4A). Mutation of the *AtC4H1* gene severely impairs the synthesis of lignin and cell wall formation, leading to slow, phloem-mediated auxin transport [30]. Certainly, many cell wall formation genes were downregulated in *ERF012*-OE seedlings (Appendix A), implying that *ERF012* expression coordinated multilayered regulation to ensure auxin decline in the root. Besides, *ERF012* expression displayed a negative effect on the root tip growth and development, which may be due to ethylene accumulation. Ethylene accumulation dramatically inhibits root elongation and promotes root hair lushness [50,51]. Overexpression of *ERF012* activated the ethylene biosynthetic genes *ACS7* and *ACS11* to cause root growth retardation and root hair increase (Figure 5A–C), which could be partially counteracted by the AVG, an ethylene biosynthetic inhibitor (Figure 5B,C). On the other hand, overexpression of *ERF012* in plants did appear to lose sensitivities to various environmental stresses such as drought, salinity and heavy metals, although some of the related genes were still regulated by *ERF012* directly or indirectly (Figure 8 and Appendix A), suggesting that *ERF012* has more effects on growth and development than on environmental stresses. Taken together, our findings provide a comprehensive evaluation of the functional versatility of the *ERF012* transcription factor in Arabidopsis growth, development and abiotic stress responses.

## 4. Materials and Methods

### 4.1. Plant Materials and Growth Conditions

For the solid medium growth, seeds were sterilized with 75% alcohol (*v*/*v*) for 1 min and then immediately immersed with 1% (*w*/*v*) NaClO for 10 min. The seeds were then washed with pure water 5 times. Then, vernalization of the washed seeds was conducted and they were incubated at 4 °C for 2 days. The 1/2 strength MS medium supplemented with 1% sucrose and 1% (*w*/*v*) gellan gum (Wako Pure Chemicals, Osaka, Japan) was used for normal conditions. The solid plates were grown in a growth chamber with a 16 h light/8 h dark cycle at 22 °C. For the IAA and AVG treatments, 5-day-old seedlings grown on a normal growth medium were transferred to the medium supplemented with IAA (1 µM) or aminoethoxyvinyl glycine (AVG) (1 μM) for another 6 days of growth. In the temperature stress and the abiotic stress experiments, the 6-day-old seedlings were treated at 4 °C or 37 °C, or with mannitol (100 mM), sodium chloride (NaCl, 100 mM), sodium arsenite (NaAsO_2_, 10 μM) or cadmium chloride (CdCl_2_, 15 μM) for another 6 days of growth, respectively. Plant growth parameters were measured by using Image J software (National Institutes of Health, and LOCI University of Wisconsin; USA), with images obtained from an Olympus SZX16 stereoscopic microscope (Olympus, Tokyo, Japan) and a digital camera.

### 4.2. Vector Generation and Plant Transformation

To construct 35S:*ERF012* transgenic plants, the open reading frame (ORF) sequences of *ERF012* were amplified by polymerase chain reaction (PCR) using specific primers (Appendix A). The *ERF012* CRISPR vector was generated according to the method of Yang [52]. To construct the p*ERF012*:GUS vector, the promoter sequence of *ERF012* was amplified by PCR using the specific primers, then fused with the *Sma* I restriction endonuclease-digested DX2181 using a 5× infusion kit (Clontech, Takara, Beijing, China). The *Agrobacterium* GV3101-mediated flower-dip method was used to generate the transgenic lines [53]. For the transcription activation assay of ERF012, the effector vectors were constructed according to the methods [45,54]. The *ERF012* CDS sequence fragment was amplified by PCR, then inserted into the binary vector GAL4DB [55] using the *EcoR* I restriction endonuclease. For the yeast one-hybrid assay, the promoter of 4CL and C4H were inserted into a pHis2 vector using *Sma* I by the 5× infusion kit (Clontech, Takara, Beijing, China), and the CDS sequences of *ERF012* were inserted into the pGADT7-rec2 vector using *Xma* I and *Xho* I with specific primers. All the specific primers are listed in Appendix A.

### 4.3. Transient Expression in Arabidopsis Protoplasts

The *Arabidopsis* protoplasts’ preparation, cotransfection and the expression analysis were carried out by the modified method, as previous described [56]. We used 3 μg of the GAL4-LUC reporter plasmid and GALDB-ERF012 effector plasmid, respectively. In addition, to normalize each transfection value, 1 μg of plasmid Ubi-Rennila LUC was used as an internal control [49]. The Dual-Luciferase Reporter Assay System was used to detect the luciferase assays through a luminescence reader (TECAN Infinite M200, Switzerland; Promega, Madison, WI, USA). The data were from at least three independent experiments with consistent results.

### 4.4. β-Glucuronidase (GUS) Histochemical Staining and Quantification of GUS Activity

Seedlings of p*ERF012*:GUS lines were incubated in a staining solution prepared using the GUS Histochemical Kit (bioshap, China, Cat. BL622A). The samples were observed under an Olympus SZX16 stereomicroscope. GUS activity was assayed according to the method [57].

### 4.5. Yeast One-Hybrid Assay

The Y1H assay was performed according to the modified method [57]. Briefly, 1000 ng pHis2-4CL or pHis2-C4H plasmids were cotransformed into the yeast strain Y187 with the 1000 ng pGADT7-rec2-ERF012 plasmid. The transformed Y187 competent cells were spotted onto a medium (SD/-Trp-Leu, Clontech, Takara, Beijing, China) and grown for 3 days at 30 °C. Subsequently, 10 μL of each diploid of the yeast cultures (OD600 = 0.1) with 1-fold, 10-fold, or 100-fold dilution was plated to solid SD/-Trp-Leu-His suppled with or without 50 mM 3-AT for 3 days at 30 °C. pGADT7-rec2-53 and pHis2-53 were the active control, and pGADT7-rec2-ERF012 and pHis2 were the negative control.

### 4.6. Anthocyanin Measurement

To analyze anthocyanin content, 0.3 mg shoots were harvested and frozen with liquid nitrogen immediately. Anthocyanins were extracted with 1 mL methanol containing 1% (*v*/*v*) hydrochloric acid and then were shaken at room temperature (50 R/min) with a shaker for 18 h. After centrifuging at 14,000 rpm for 10 min, 0.4 mL of the suspension was transferred to 0.6 mL methanol containing 1% (*v*/*v*) hydrochloric acid in a 1.5 mL tube. The absorbance of diluted supernatants was measured using a microplate reader (TECAN Infinite M200, Switzerland) at 530–675 nm. The anthocyanin calculation formula is QAnthocyanins = (ODA530-0.25×ODA657)/sample fresh weight (g).

### 4.7. Quantification Chlorophyll Concentration

The quantification of chlorophyll concentration was performed following the modified protocol that was described by Lichtenthaler and Buschmann [58]. About 100 mg fresh weight of leaves in seedlings was ground with 2 mL of 80% isopropyl alcohol (*v*/*v*), which contained 10 mg CaCO_3_. After centrifuging at 12,000 rpm for 10 min at 4 °C, then the suspension was transferred to a 1.5 mL tube. The diluted supernatants were measured using a microplate reader (TECAN Infinite M200, Switzerland) at the absorbance wavelengths of 663.2 and 646.8 nm. The concentrations for chlorophyll a (Ca) and chlorophyll b (Cb) were calculated according to the formula: Chlorophyll = (20.29A646.8 + 8.05 A663.2) × *v*/*w* × 1000.

### 4.8. RNA Extraction and qRT-PCR ANALYSIS

The shoots and roots of the Col-0 and OE *ERF012* grown under indicated growth conditions were sampled with three independent biological replicates for the analysis of the *ERF012* expression. Total RNA was extracted using the Trizol Reagent (Invitrogen, Carlsbad, CA, USA) according to the manufacturer’s instructions. Then, 1 μg total RNA in a 10 μL reaction volume was used for reverse transcription into a single cDNA. Once the reverse transcription reaction was finished, the cDNA was diluted 20 times. A quantitative reverse transcription-PCR (qRT-PCR) for detecting the relative expression of genes was performed using the SYBR Green Real-Time PCR Master Mix Kit (TOYOBO, Osaka, Japan) and the CFX96^TM^ Real-Time PCR Detection System (Bio-Rad, Hercules, CA, USA). The composition (4 μL cDNA templates, 0.2 μL 10 μM primers, 5 μL SYBR Green mix, 0.4 μL sterile ddH2O) were based on 10 μL reaction volume per well for 96-well plates. *Arabidopsis* actin and UBQ5 were used as two endogenous controls for sample normalization. The gene-specific primer sequences used are listed in Appendix A. Expression data were normalized with the expression level of the AtActin and AtUBQ5 by the 2^−ΔΔCT^ method [59].

### 4.9. Statistical Analysis

Statistical analysis was performed using SPSS16.0 for Windows software (SPSS 249 Inc., Chicago, IL, USA). Significant differences among treatments were analyzed by analysis of variance and Duncan’s test at the *p* < 0.05 level, and Student’s *t*-test at the * *p* < 0.05, ** *p* < 0.01 and *** *p* < 0.001 level.

## Figures and Tables

**Figure 1 ijms-23-06841-f001:**
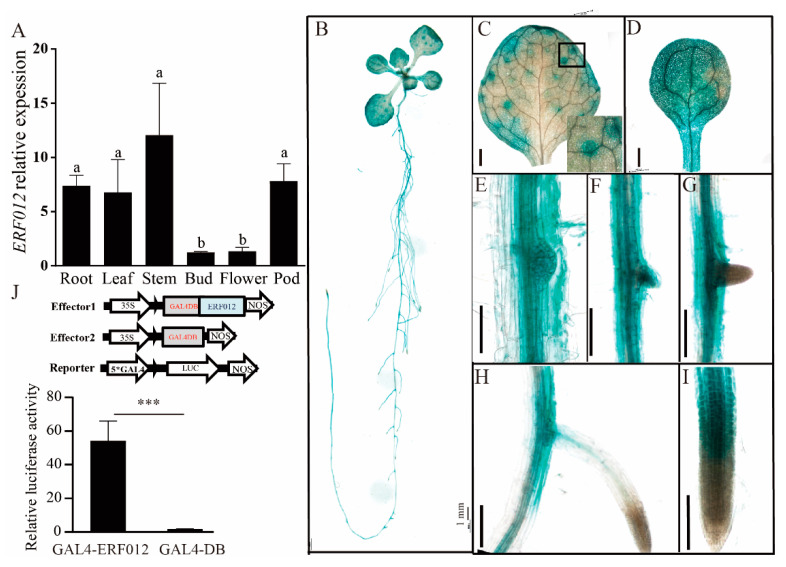
Tissue expression and transactivation assay of *Arabidopsis ERF012*. (**A**) Expression levels of *ERF012* in different tissues. Thirty-five-day-old *Arabidopsis* was used for qRT-PCR analysis. Letters indicate significant differences between different tissues: Duncan’s test (*p* < 0.05). (**B**–**I**) Expression pattern analysis of *ERF012* in seedlings. Twelve-day-old seedling expressing p*ERF012*:GUS was used. A close-up was indicated by a black frame in (**C**). Scale bar = 100 μm in (**C**–**I**). (**J**) Transactivation assay of *ERF012* genes in protoplast. GAL4-ERF012 is the effector and GAL4-LUC is the reporter. GAL4-DB is the control. The values are means ± SD, *n* = 5. Asterisks indicate significant differences between different treatments: Student’s *t*-test, *** *p* < 0.001.

**Figure 2 ijms-23-06841-f002:**
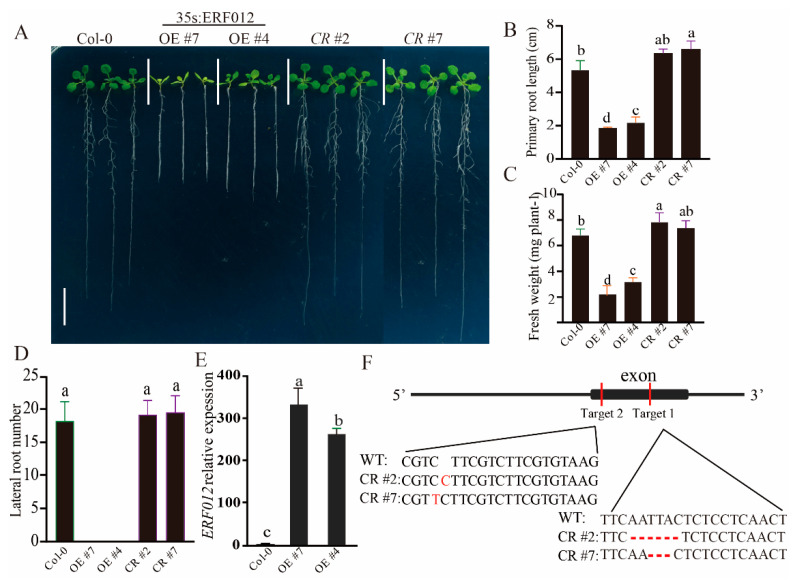
ERF012 regulates plant growth and development. (**A**) Phenotypical comparison of Col-0, *ERF012* overexpression in plants and *erf012* mutants. Twelve-day-old seedlings grown on 1/2 MS solid medium were imaged, scale bar = 1 cm. (**B**–**D**) The primary root length, shoot fresh weight and the lateral root number were statistically measured in (**A**), *n* = 18 plants. (**E**) Validation of *ERF012*-OE plants by qRT-PCR analysis; *n* = 3 pools, with about 20 plant roots per pool. (**F**) Mutation information of *ERF012* in its exon sequence established by a CRISPR/Cas9 system. Letters in (**B**–**E**) indicate significant differences between different plants: Duncan’s test (*p* < 0.05).

**Figure 3 ijms-23-06841-f003:**
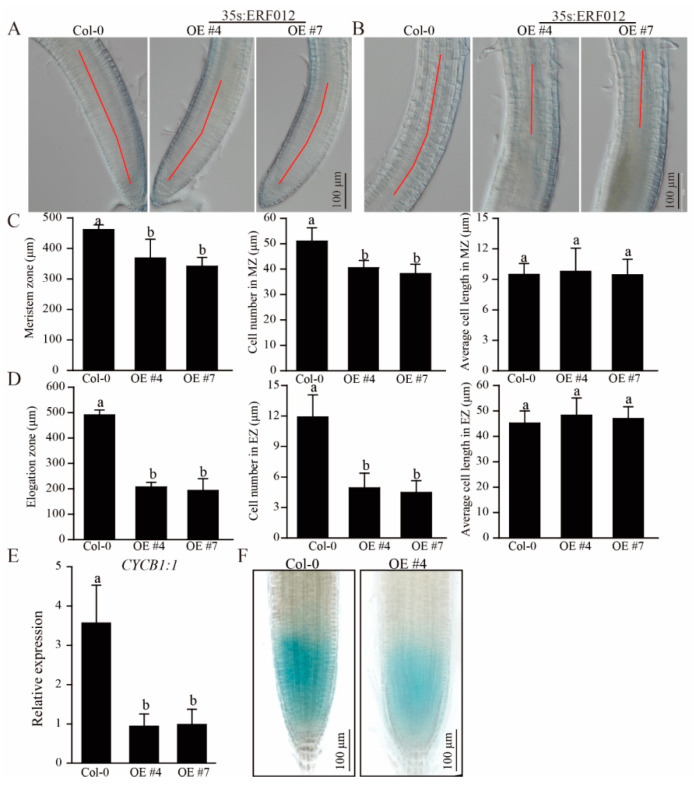
ERF012 involves root cell growth and division. (**A**,**B**) Root tip images of Col-0 and *ERF012*-OE lines. The red lines indicate the meristem zone (**A**) and elongation zone (**B**). (**C**,**D**) The statistical calculation of zone length, cell number and average cell length in root tips of Col-0 and *ERF012*-OE lines; *n* = 12 roots. Values represent means ± SD. (**E**) The expression of *CYCB1:1* in the roots of Col-0 and *ERF012*-OE lines. Twelve-day-old seedlings were used. Values represent means ± SD, *n* = 3 pools, with about 20 plant roots per pool. (**F**) Transcription activity analysis of CYCB1;1 in Col-0 and *ERF012*-OE backgrounds expressing the p*CYCB1:1*:GUS. Letters in (**C**–**E**) indicate significant differences between different plants: Duncan’s test (*p* < 0.05).

**Figure 4 ijms-23-06841-f004:**
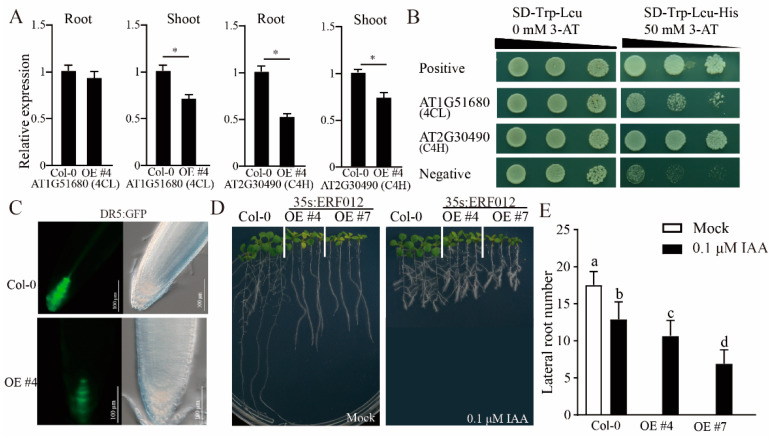
Overexpression of *ERF012* regulates lateral root formation. (**A**) The expression of *At4CL* and *AtC4H* in 12-day-old Col-0 and *ERF012*-overexpressing lines (OE #4). Values represent means ± SD, *n* = 3 pools, with about 20 plant roots per pool. Asterisks indicate significant differences between different plants: Student’s *t*-test, * *p* < 0.05. (**B**) *ERF012* binds to the promoter of the *At4CL* and *AtC4H*. The promoter sequences of the *4CL* and *C4H* were subcloned into the pHis2 vector and the *ERF012* CDS sequence was subcloned into the pGADT7-rec2 vector for yeast one-hybrid assay. (**C**) The images of DR5:GFP in root tips of Col-0 and *ERF012*-OE lines (OE#4). (**D**) Exogenous application of IAA (0.1 μM) can effectively restore the lateral root formation. The 5-day-old seedlings were transferred to the medium supplemented with or without IAA for 6 days of growth. (**E**) The lateral root number was statistically calculated in figure (**D**). Values represent means ± SD, *n* = 12. Letters indicate significant differences between different plants: Duncan’s test (*p* < 0.05).

**Figure 5 ijms-23-06841-f005:**
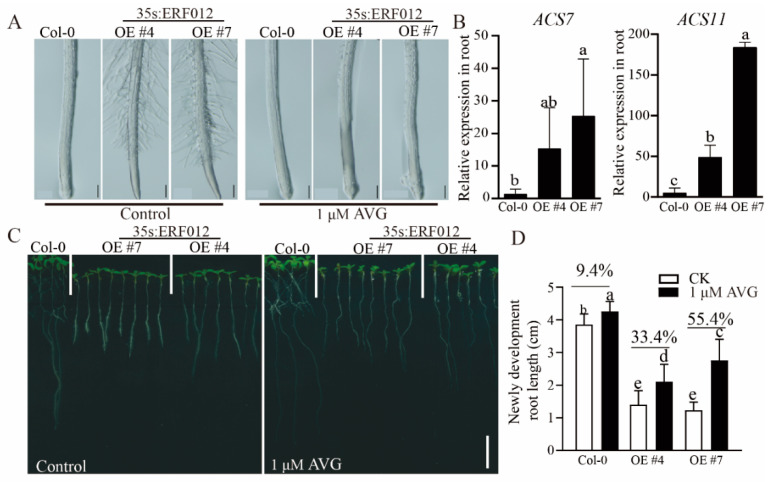
ERF012 activates ethylene biosynthesis in the root and regulates root tip growth. (**A**) Ethylene biosynthetic inhibitor AVG (1 μM) treatment downregulated root hair density in *ERF012*-OE lines. Scale bar = 100 μm. (**B**) Expression levels of ethylene precursors and biosynthetic genes *ACS7* and *ACS11* in Col-0 and *ERF012*-OE lines. Values represent means ± SD, *n* = 3 pools, with about 20 plant roots per pool. (**C**) Ethylene biosynthetic inhibitor AVG (1 μM) treatment restored root growth in *ERF012*-OE lines. Five-day-old seedlings were transferred to the medium supplemented with or without IAA for 6 days of growth. Scale bar = 1 mm. (**D**) Statistical calculation of the newly developed root length treated with or without AVG (1 μM) in figure (**C**). Values represent means ± SD, *n* = 16. Letters indicate significant differences between different plants: Duncan’s test (*p* < 0.05).

**Figure 6 ijms-23-06841-f006:**
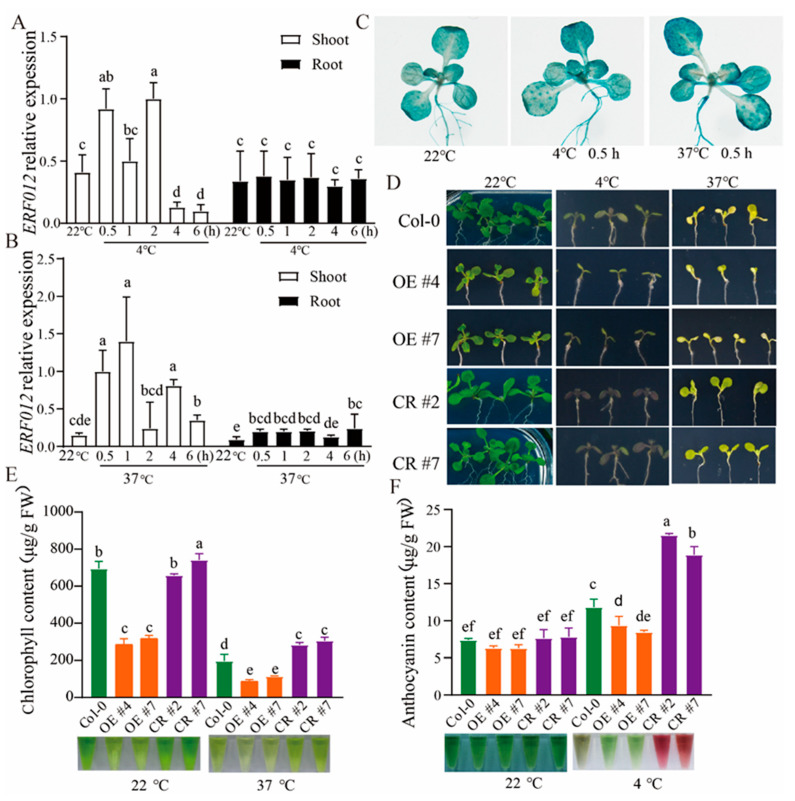
ERF012 responds to temperature dynamics. (**A**,**B**) Time-course analysis of *ERF012* expression exposed to 4 °C or 37 °C for 0.5 h, 1 h, 2 h, 4 h, 6 h. Twelve-day-old seedlings were used for qRT-PCR. Values represent means ± SD, *n* = 3 pools, with 12 seedlings per pool. Letters indicate significant differences between different plants: Duncan’s test (*p* < 0.05). (**C**) GUS staining of p*ERF012*:GUS seedlings exposed to C for 0.5 h. (**D**) The shoot phenotype of 6-day-old Col-0, *ERF012*-OE lines and *erf012* mutant lines at 4 °C or 37 °C. Six-day-old seedlings grown at 22 °C were transferred to 4 °C or 37 °C for 6 days of growth. (**E**,**F**) The chlorophyll content (37 °C treatment) and anthocyanin content (4 °C treatment) of all plants were statistically calculated. Values represent means ± SD, *n* = 4 pools, with 5 shoots per pool. Letters indicate significant differences between different plants: Duncan’s test (*p* < 0.05).

**Figure 7 ijms-23-06841-f007:**
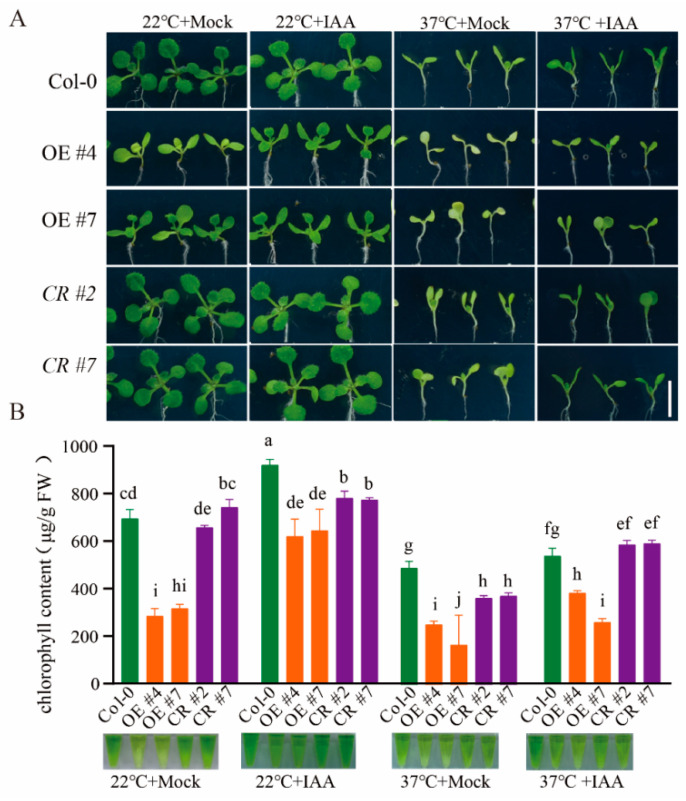
Auxin significantly relieves ERF012-mediated leaf senescence under temperature dynamics. (**A**) Phenotypical comparison of leaf senescence of Col-0, *ERF012*-OE lines and *erf012* mutants at 22 °C or 37 °C. Six-day-old seedlings grown on 1/2 MS medium at 22 °C or 37 °C were transferred to the medium with or without 1 μM IAA for 6 days of growth. (**B**) The chlorophyll content of all plants was statistically calculated. Values represent means ± SD, *n* = 4 pools, with 5 shoots per pool. Letters indicate significant differences between different plants: Duncan’s test (*p* < 0.05).

**Figure 8 ijms-23-06841-f008:**
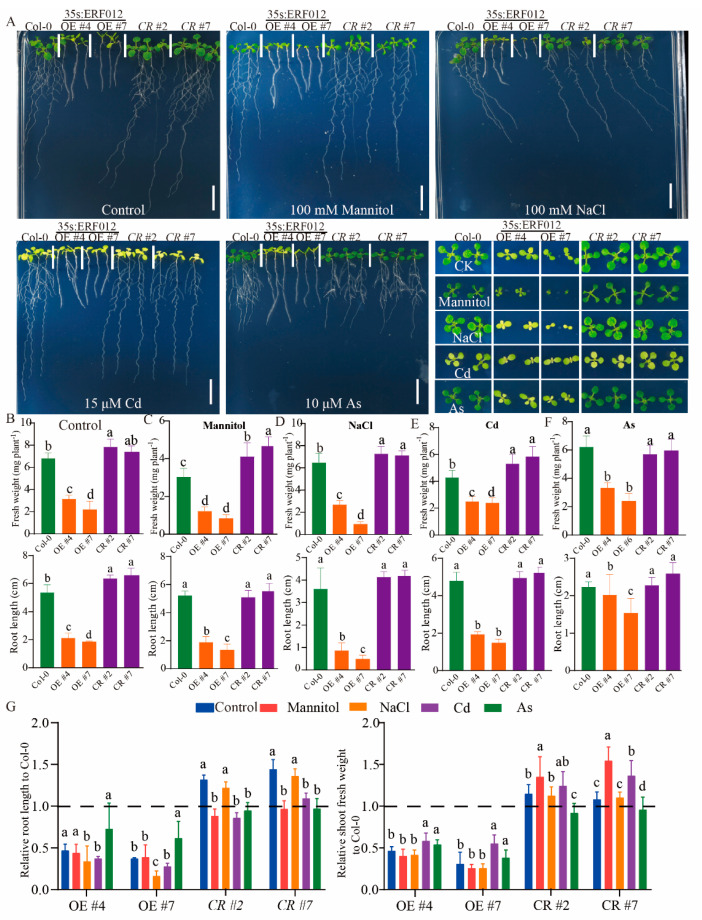
ERF012 reduces plant sensitivity to various abiotic stresses. (**A**) The phenotypical comparison of Col-0, *ERF012*-OE lines and *erf012* mutant lines. Six-day-old seedlings grown on 1/2 MS medium were transferred to the medium with 100 mM mannitol, 100 mM NaCl, 10 μM arsenite and 15 μM CdCl_2_ for 6 days of growth. (**B**–**F**) The shoot fresh weight and primary root length were statistically calculated. (**G**) The relative primary root length of Col-0 and relative shoot fresh weight of Col-0 under each stress were statistically calculated. (**B**,**G**) Values represent means ± SD, *n* = 4 pools, with 5 shoots per pool. Letters indicate significant differences between different plants: Duncan’s test (*p* < 0.05). Dotted lines (**G**) indicate the relative Col-0 root length and relative Col-0 shoot fresh grown in normal conditions.

## Data Availability

Not applicable.

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
