# Peer review of "Arabidopsis ERF012 Is a Versatile Regulator of Plant Growth, Development and Abiotic Stress Responses"

_ijms, 2022, doi:10.3390/ijms23126841_

Round 1

Reviewer 1 Report

In this manuscript, Huang and co-authors reported that ERF012 transcription factor negatively regulated root growth and development, by inhibiting lateral root emergence through the regulation of plant hormones. IAA and ethylene are key factors which regulated root growth in ERF012 OE plants. The authors also linked ERF012 with abiotic stress resistance.

My comments,

1, in abstract, it was mentioned that ERF012 overexpression accelerated leaf senescence. I am wondering the yellow leaf was due to senescence or nutrient stress. To my understand, it more looks like the ERF012 leaves suffered nutrient deficiency, which was due to the bad growth of root.

2, the authors should be careful about the figure legends. All of them should be improved.

“letters indicate significant differences between different treatments”, actually it is not the comparison of different treatments in all figures.

Figure 1, scale bars are needed in B-I. It is not clear for me what does it mean of Figure 1J, it will be helpful to explain more in the legend and/or add schematic draw to explain. I prefer to remove “tissue specific”.

Figure 2, the mutation was confirmed in root or leaf ? it will be nice to quantify the lateral root number between WT and ERF deregulated plants.

Figure 3, “ERF012 is involved in root cell growth and division”. At the end of the legend, what does it mean “n=10”.

Figure 4, 0.1uM ABA or 1uM ABA was applied? Explain “ND” in Figure 4E. In figure 4E Y-axis, should be “lateral root number”.

Figure 5, why no root hairs in Col-0 plants? Replace all “CK” with “Control”. Explain what is “newly development root length”.

Figure 7, why auxin was tested with temperature treatments.

3, line 219, for ERF OE or Col-0 shoots?

Author Response

Response to Reviewers

Reviewer 1:

Comments and Suggestions for Authors

In this manuscript, Huang and co-authors reported that ERF012 transcription factor negatively regulated root growth and development, by inhibiting lateral root emergence through the regulation of plant hormones. IAA and ethylene are key factors which regulated root growth in ERF012 OE plants. The authors also linked ERF012 with abiotic stress resistance.

Response:We appreciate your positive and constructive comments on our manuscript, and for allowing us to revise it!

My comments,

1, in abstract, it was mentioned that ERF012 overexpression accelerated leaf senescence. I am wondering the yellow leaf was due to senescence or nutrient stress. To my understand, it more looks like the ERF012 leaves suffered nutrient deficiency, which was due to the bad growth of root.

Response: Thank you for the question! According to your comment, we measured the nutrient level. The results were shown as Supplemental Figure 3. We found that compared to the Col-0, the P and Zn were decreased in the OE ERF012 shoot, the others were comparable or higher. P deficiency usually causes purple leaves. The Zn reduction in ERF012 OE shoots might be an incentive for leaf senescence. We include this content in the discussion section.

 Supplemental Figure Figure 3: The nutrition in Col-0 and ERF012 OE lines. The Col-0 and ERF012 OE lines were grown in hydropic solution for 35 d. The shoots and roots were harvested and dried for nutrition detection.

2, the authors should be careful about the figure legends. All of them should be improved.

“letters indicate significant differences between different treatments”, actually it is not the comparison of different treatments in all figures.

Response: Thanks very much for your kind advice! According to your suggestion, we revised the obscure describe in every figure lenged as below:

Figure 1: Letters indicate significant differences between different tissues: Ducan’s test (p<0.05). 

Figure 2: Letters (B-D) indicate significant differences between different plants: Ducan’s test (p<0.05).

Figure 3: Letters (C-E) indicate significant differences between different plants: Ducan’s test (p<0.05).

Figure 4: Asterisks (C-E) indicate significant differences between different plants: Student’s t-test: *, P < 0.05.

Letters indicate significant differences between different plants: Ducan’s test (p<0.05).

Figure 5: Letters indicate significant differences between different plants: Ducan’s test (p<0.05).

Figure 6: Letters indicate significant differences between different plants: Ducan’s test (p<0.05).

Figure 8: Letters indicate significant differences between different plants: Ducan’s test (p<0.05).

Figure 1, scale bars are needed in B-I. It is not clear for me what does it mean of Figure 1J, it will be helpful to explain more in the legend and/or add schematic draw to explain. I prefer to remove “tissue specific”.

Response: Thanks very much for your kind advice! We added the scale bars in Figure 1B-I, and draw a diagram to show the experiment of transcriptional activation. In addition, according to your suggestion, we revised the “Tissue-specific expression and transactivation assay of Arabidopsis ERF012” into“Tissue expression and transactivation assay of Arabidopsis ERF012” in Figure 1 legends.

Figure 2, the mutation was confirmed in root or leaf ? it will be nice to quantify the lateral root number between WT and ERF deregulated plants.

Response: Thank you for the question! We did the ERF012 expression in the root of ERF012 OE plants and Col-0, but not in the erf012 mutants. The ERF012 mutants were generated by CRISPR-Cas9, which causes mutation in exon region of genome, thus would not change the mRNA level. The edited information of cDNA was shown in Figure 2F.

In addition, according to your suggestion, we add the lateral root number data in the Figure 2D, and added the sentence “As shown in Figure 2A, D, ERF012 OE lines obviously had lower lateral root number than Col-0 and erf012 mutants.”in therevised manuscript.   

Figure 3, “ERF012 is involved in root cell growth and division”. At the end of the legend, what does it mean “n=10”

Response: Thank you for the question! This is our mistake and should be deleted.

Figure 4, 0.1uM ABA or 1uM ABA was applied? Explain “ND” in Figure 4E. In figure 4E Y-axis, should be “lateral root number”.

Response: Thank you for the question! We used the 0.1 μM IAA in the experiment. Because the OE ERF012 lines are almost without root hairs, we named as “ND”. This is not accurate. We have deleted it. In addition, according to your suggestion, we revised the “later root number” into the “lateral root number” in Figure 4.

Figure 5, why no root hairs in Col-0 plants? Replace all “CK” with “Control”. Explain what is “newly development root length”.

Response: Thank you for the question! In the normal condition, the root tip is relative longer and the root hairs are gradually emerging along the root. Figure 5 showed the root tip including some immature/elongating root hairs, therefore, only a visible small bump on the cell surface can be observed. And, AVG further inhibits this process.

Thanks very much for your kind advice! According to your suggestion, we replaced all “CK” with “Control”.

 “newly development root length” means the value of primary development after treatment. We revised it as “ newly developed root length”.

Figure 7, why auxin was tested with temperature treatments.

Response: Thank you for the question! As explained in the discussion section, the leaf senescence could be caused by various stimuli. At the normal conditions, no environmental stresses were added. Thus, the internal factors are the reason. Based on our nutrient analysis, most nutrients were comparable or a little bit higher in the ERF OE shoots than the Col-0 except for the P and Zn. Although we cannot exclude the possibility of Zn deficiency on leaf senescence, the ERF012 overexpression inhibits auxin level in root is a solid result, and some studies have been implicated the importance of IAA in high-temperature stress. Therefore, the auxin was tested with temperature treatments. 

3, line 219, for ERF OE or Col-0 shoots?

 Response: Thanks very much for your kind advice! According to your suggestion, we revised ERF012 OE as Col-0 in line 219 of the manuscript.

Reviewer 2 Report

The reviewed manuscript by Huang et al. entitled “Arabidopsis ERF012 is a versatile regulator of plant growth and development and abiotic stress responses” is an interesting study. The authors have collected a unique dataset using cutting-edge methodology. The paper is generally well written and structured. Here authors studied the role of ERF012 transcription factors on plant growth and development as well as abiotic stress tolerance particularly drought, heat, and heavy metal tolerance.

 However, I have general comments following;

a)     Authors should add more specific functions of ERF012 regarding abiotic stress (drought, HM, heat) tolerance and interactions with other genes.

b)     Similarly, the discussion part is short with no details of previous findings/evidences of such transcription factors, it looks similar to the results part.

c)     Please add little bit of details on the Real-Time PCR program

Finally, authors are suggested to correct some grammatical errors

Author Response

Response to Reviewers

Reviewer 2

Comments and Suggestions for Authors

The reviewed manuscript by Huang et al. entitled “Arabidopsis ERF012 is a versatile regulator of plant growth and development and abiotic stress responses” is an interesting study. The authors have collected a unique dataset using cutting-edge methodology. The paper is generally well written and structured. Here authors studied the role of ERF012 transcription factors on plant growth and development as well as abiotic stress tolerance particularly drought, heat, and heavy metal tolerance.

Response: We appreciate your positive and constructive comments on our manuscript, and for allowing us to revise it!

 However, I have general comments following;

  1. Authors should add more specific functions of ERF012 regarding abiotic stress (drought, HM, heat) tolerance and interactions with other genes.

Response: Thank you very much. It is a good suggestion. We agree with this that ERF012 should be multiple regulatory factors and interact with various stresses-related genes because of the significant expression levels of PCS2, GSH2, NRAMP3/4 and so on (supplemental Figure2). However, we cannot obtain these results within a short time (10 days). On the other hand, as disused in M.S, the growth control by ERF012 appears to play the epistatic effect covering other abiotic stress responses.

  1. Similarly, the discussion part is short with no details of previous findings/evidences of such transcription factors, it looks similar to the results part.

Response: Thank you very much. We expanded the discussion section that included many details of previous findings of ERF TF. Please check the changes.

  1. Please add little bit of details on the Real-Time PCR program

Response: Thank you very much. We added the details on the Real-Time PCR program in the methods section.

Finally, authors are suggested to correct some grammatical errors

Response: Thank you very much. We get help from a language expert and the revised M.S with fewer grammatical errors.

Round 2

Reviewer 1 Report

Thanks for the revision! I have no more comments.